# Optical Protective Window Design and Material Selection Issues in the Multi-Sensor Electro-Optical Surveillance Systems

**DOI:** 10.3390/s23052784

**Published:** 2023-03-03

**Authors:** Saša Vujić, Dragana Perić, Branko Livada, Miloš Radisavljević, Dragan Domazet

**Affiliations:** 1Vlatacom Institute, 11070 Belgrade, Serbia; 2Faculty of Information Technologies, Doctoral Studies in Software Engineering, Belgrade Metropolitan University, 11158 Belgrade, Serbia

**Keywords:** surveillance systems, multi-sensor imaging systems, electro-optical imaging sensors, infrared imaging sensors, optical protective window, optical materials, optical coatings, opto-mechanical design

## Abstract

Multi-sensor imaging systems have a very important role and wide applications in surveillance and security systems. In many applications, it is necessary to use an optical protective window as an optical interface connecting the imaging sensor and object of interest’s space; meanwhile an imaging sensor is mounted in a protective enclosure, providing separation from environmental conditions. Optical windows are often used in various optical and electro-optical systems, fulfilling different sometimes very unusual tasks. There are lots of examples in the literature that define optical window design for targeted applications. Through analysis of the various effects that follow optical window application in connection with imaging systems, we have suggested a simplified methodology and practical recommendation for how to define optical protective window specifications in multi-sensor imaging systems, using a system engineering approach. In addition, we have provided initial set of data and simplified calculation tools that can be used in initial analysis to provide proper window material selection and definition of the specifications of optical protective windows in multi-sensor systems. It is shown that although the optical window design seems as a simple task, it requires serious multidisciplinary approach.

## 1. Introduction

Multi-sensor imaging systems have a very important role and wide application areas in long-range surveillance and security systems [1,2,3,4,5]. They are adaptable modular systems with multi-spectral imaging sensors used for observation, whose purpose is to provide data regarding objects of interest, i.e., targets inside the field of regard (FOR). Surveillance systems use cameras as a key component in the imaging process. In the majority of applications, the camera needs to be isolated from the environment using a transparent optical window. The purpose of an optical protective window is to provide a clear optical aperture to transmit the desired radiation and to keep the two environments separated.

An optical protective window is used as optical interface connecting the imaging sensor and object of interest space; thus, proper selection of a window’s optical performance is fundamental to guarantee that imaging channel data will not be degraded due to the window of the multi-sensor surveillance system.

Optical protective windows are usually built as plane parallel plates of transparent optical material that allow undisturbed application of the imaging sensor while protecting the given system from dust and debris and providing environmental protection and vacuum seals. In addition, optical protective windows are used to protect sensitive and expensive imaging sensors from any harmful actions that could happen during system usage. The protective window design should also allow for easy window replacement. Because of this, optical protective windows are often called “sacrificing glass”.

Optical windows can be designed in other more complicated shapes such as domes, segmented windows and conical sections. In addition to the fact that windows provide mechanical protection for optical systems, they are also important optical elements of these systems. Even though the windows can be simple plane parallel plates, extreme environmental conditions such as pressure and temperature variations can cause the windows to distort incoming optical signal, thereby diminishing the performance of the optical sensor.

One of the most demanding applications of the protective window is the infrared (IR) dome in the missile homing head, wherein the dome is a part of the aerodynamic envelope and exposed to extreme mechanical and thermal loads; therefore, all effects must be treated seriously [6]. The design of optical windows is described in several opto-mechanical books [7,8,9]. Optical window shapes and their potential deformations during exploitation can cause serious image degradation in aerospace systems [10,11,12,13,14,15]. This leads to specialized and expensive technical solutions. The complexity of the related solution depends mostly on the specific application, and always requires additional research and developmental efforts from the multidisciplinary teams involved. The most illustrative examples of this are so-called aero-optical effects, which are not present in the case of the surveillance systems but show all physical sources of the potential image degradation caused by protective window. We reviewed the aero-optical effect in the Appendix A, wishing to refer to additional sources of information that can be used for detailed analysis.

It seems that in multi-sensor surveillance systems, environmentally caused image degradations are not critical, but in long-range systems, the high resolution of the imaging sensors adds new requirements for window material and design quality.

New challenges in defining optical protective windows’ requirements are bound to come with the application of the multi-sensor systems in autonomous driving vehicles and with the application of machine vision systems in agriculture, wherein window cleaning can be very important to provide constant readiness.

One of the goals of this article is to present a condensed review of the effects that can cause image degradation. We use scientific results generated in the more complicated application cases, and define simplified guidelines for optical protective window design for application in stationary surveillance systems.

For optical protective windows in the multi-sensor surveillance system containing more electro-optical (EO) and infrared (IR) imaging channels, the process of defining their requirements is treated as a system design challenge. The potential impact of the window’s properties and assembly methods on EO/IR imaging systems’ performance and sensitivity will be discussed following selected practical experience and laboratory-based simulations.

First, we explore the multi-sensor systems’ generalized design and describe the role of the optical protective windows in such systems. After that, we describe the most important effects that could appear in various applications, and their influence on defining the requirements of window design. Additionally, we show some results obtained previously to illustrate how window design specifications influence the final system quality, and what can go wrong in the exploitation period.

The design of optical protective windows seems as a simple task, but there are several factors a designer must be aware of when designing a window for an optical system (e.g., mechanical strength, transmission range, environmental durability, optical properties and available coatings). Optical protective window materials and coating choices are limited, so they should be carefully selected during multi-sensor system design requirements’ development. The design requirements also lead to the optimization of numerous mechanical, optical, material, and electrical parameters.

The key objective of this paper is to provide a review of the most important effects of optical protective windows’ implementation in multisensory surveillance systems, and refer to the other papers with more detailed treatment. Following the surveillance systems’ specificities, we present a simplified methodology and procedures which may assist in defining initial design requirements, and which proves to be sufficient in most cases.

The work presented in this paper provides a simplified methodology and practical recommendations for an easier systems engineering approach to defining optical protective windows’ specifications for multi-sensor imaging systems. This work has resulted from a system engineering approach, through the analysis of various effects that are common to optical window application in relation to imaging systems.

## 2. Optical Protective Window Role in Multi-Sensor Surveillance Systems

Multi-sensor surveillance systems are adaptable modular systems composed of several imaging sensors mounted at the sensor head and controlled remotely [16]. The design of multi-sensor surveillance systems depends on application of many multidisciplinary scientific fields supporting the required use case scenarios.

A typical multi-sensor surveillance system is shown in Figure 1a. The multi-sensor system has the following key components:

**The Imaging Group**, which contains several imaging channels providing images of the space of interest. The simplified structure of one imaging channel is shown in Figure 1b. The system is composed using some or all of the elements listed below.

Daylight (low light) video imaging sensorIR imagers—(short wave infrared—SWIR, mid-wave infrared—MWIR, long wave infrared—LWIR) imaging sensorsLIDAR sensorUV imaging sensor

**The Position Sensing and Control group**, which provides the sensor head’s position and orientation in the space, and collects position and orientation related data.

Imaging sensors’ video signals and position sensors’ data are processed and integrated using an EO system built in computer command and control software packages. Advanced solutions for image enhancement, image stabilization and sensor data fusion may be optionally added.

The basic purpose of an optical protective window in a complex electro-optical system is to provide a clear optical aperture and to keep the optical sensor enclosed and separated from outside environmental influences. The window should be designed to provide minimal degradation of the optical sensor performance and to protect sensitive parts from harsh environmental impact. This means that the window will be exposed to all environmental influences, and because of that, window design and sealing methods applied should be rigid enough to minimize mechanical and thermal distortions.

The desirable characteristics of an optical window are listed below.

○Low absorption of transmitted light○Low reflection of light incident on the surfaces of the window○Low refraction (or bending) of the transmitted light rays○Low scattering to minimize stray light influence and contrast degradation○Minimum distortion of the transmitted beam due to imperfections in the optical material or the surface finish, keeping wave front undisturbed and image sharpness unchanged○Durable and not susceptible to damage, having an ability to withstand degradation from environmental causes. This may include strength, resistance to water or temperature, etc.

With the use of a protective window in the system, we introduce a new optical element which will influence the imaging sensor’s overall optical performance. The insertion of an optical protective window on the image optical path may introduce several potential problems, including

(a)reduced transmission resulting from the reflection, absorption, and scattering of the window material, causing reduction of the operational range of the sensors [17],(b)contrast reduction caused by lack of control of the reflected and scattered energy in the window material,(c)a further increase in scatter or reduction of transmission by environmental effects such as erosion, or accumulated dust and dirt on the window surface,(d)an increase in the wave front aberrations due to the window shape, the window clear optical opening, and the positioning of the associated imaging system,(e)further aberrations induced in the window, caused by the mechanical and thermal effects during exploitation.

The typical protective window application and key parameters’ definitions are illustrated in Figure 2, wherein a circular plan parallel window is shown, together with the key design parameters listed below.

T_W_—window thickness [mm]

D_W_—window diameter [mm]

D_SA_—lens input aperture [mm]

D_OW_—window optically clear area [mm]

W_M_—window mounting area width [mm]

Z_W_—window–lens distance [mm]

Θ_FOV_—imaging sensor maximal field of view [degrees]

Θ_tilt_—window tilt angle [degrees]

In the case of the plane parallel plate, there are several key design parameters that should be defined during the design process:(a)Window material and coatings;(b)Window thickness;(c)Window position and tilt to sensor optical axes; and(d)Window shape and sealing technique.

In critical applications, the window impact will require more precise study, which then should be included as an integral part of the lens design analyses. All influences and processes, as illustrated in Figure 3, should be addressed and analyzed. The most common window shape is a single flat sheet of transmitting material. If perfectly flat and perfectly optically homogeneous, it adds no aberrations to subsequent imaging of an object at infinity, but it may limit the field of view. For close objects, a focal shift and other aberrations may be added to subsequent imagery.

Accelerations, pressure differentials, other mechanical forces and thermal perturbations will affect both surface flatness and optical homogeneity.

Thermal effects can generate the following simultaneous effects on the protective window optical properties: (a) temperature dependence of the optical window material optical properties, (b) non-uniform temperature distribution over window opening, and (c) thermally induced shape changes.

Excellent thermal optical performances of the optical window are fundamental to guarantee that the imaging system will operate normally. In order to decrease the potential influence of the deformation of the window surface caused by thermal stress, an athermal design to decrease the thermal stress of optical window is an additional requirement of the selected window mounting and sealing method.

Derivation of the optical protective window requirements and definition of the protective window key design requirements should be carried out after an integrated multidisciplinary analysis [18,19,20] which has been adapted to the anticipated multi-sensor’s real application environment. While the influencing effects (as shown in Figure 3) are numerous, for most applications, some of these influences may be neglected and simplified analysis can be performed.

### 2.1. Optical Effects

Optical phenomena in the window material [21] are extremely important in the case of window application in laser systems. In any case of application of the protective window, we expect highly transparent window materials, meaning a window material should transmit light without noticeable effect on either the light beam or the window itself. We are basically concerned with the fundamental optical phenomena of absorption, refraction, and scattering in window materials, such basic properties as the index of refraction, the temperature derivative of the refractive index, dn/dT, and such window material mechanical properties as elasticity and hardness.

Additionally, we should be concerned with additional optical effects induced by external thermal and mechanical stress influencing the optical beam; thermally induced aberrations due to index of refraction temperature dependence and non-uniform heating; thermally induced birefringence in windows; and elasto-optic effects and material surface shape change due to mechanical stress, etc. In addition, we should be concerned with material mechanical properties’ ability to provide rigidity of design.

Protective optical window material properties are carefully reviewed [22,23,24,25,26], and should be used as a starting point in the window requirement derivation process. A short review of the selected optical material basic properties is presented in Table A1 in Appendix A. The specificities of the optical window, such as the optical component [13,27], should be considered during analysis of its influence in the optical part of the imaging channel.

The surface quality of an optical window is determined by evaluation of surface imperfections that may be caused during manufacturing or handling. These defects typically cause small reductions in throughput and small increases in scattered light, which have little to no adverse effect on the overall system performance in most imaging or light gathering applications. However, some surfaces are more sensitive to these defects, such as surfaces at image planes, because surface defects are in focus. Windows used with high power level lasers are also sensitive to surface defects because they can cause increased absorption of energy and damage the window.

Surface quality is often described by the scratch-dig specification in the U.S. Standard MIL-PRF-13830B [28]. The scratch designation is determined by comparing the scratches on a surface to a set of standard scratches under controlled lighting conditions. Additionally, the window surface regularity (e.g., roughness) and surface parallelism are also important and should be defined because they could introduce additional aberration and wave front deformation, which could be important in the case of long-range surveillance systems.

#### 2.1.1. Spectral Sensitivity Range

Imaging sensor spectral sensitivity defines basic window optical material selection. Electro-optical and infrared imagers are designed for limited spectral sensitivity related to particular spectral regions, as illustrated in Figure 4. These spectral regions are well suited to the so-called atmospheric transmission windows. E/O and IR imagers usually use a single spectral band, but in some cases, they are sensitive in two or more spectral bands (multi spectral imagers), or several imagers may share the same window.

The imager’s spectral sensitivity band is a starting point for optical protective window spectral transmission band definition and window optical material initial selection.

#### 2.1.2. Transmission, Reflection and Scattering

The interaction of optical radiation with the optical protective window, as illustrated in Figure 5, involves several basic processes: reflection from window boundary surfaces, absorption of radiation in the window material and scattering of radiation. All these processes affect transmission of the radiation (transmission losses), but also can affect the generated image quality. The most critical is surface reflection contributing to stray light distribution and ghost image generation.

Radiation scattering and absorption in the window material contributes to a reduction in the imager’s sensitivity (transmission losses) and image contrast reduction. These effects can be minimized using high-quality (optical grade) base material.

Window surface reflection could contribute to the stray light spreading through the imager’s lens system, causing ghost images formation and/or image contrast reduction. The reflection, r, from the window boundary surface (Fresnel reflection losses), depends on optical material index of refraction, n:(1)$$r={\left(\frac{n-1}{n+1}\right)}^{2}$$

Surface reflection is very high for materials with a high index of refraction. In order to reduce surface reflection, various techniques are applied to modify component surface [29] using different approaches. The simplest technique is to use a coating of material with lower index of refraction and to define the thickness of coating which will provide optical path (phase) difference sufficient for destructive interference between radiation reflected from the coating to air boundary and from the coating to surface boundary. The spectral reflection band and reflection value optimization is achieved using multilayer thin film coatings. Engineered surface structures (Moth’s eye) have turned out to be an effective alternative to thin-film anti-reflective (AR) coatings in many infrared and visible-band applications in which durability, radiation resistance, wide viewing angle, or broad band performance are critical.

Anti-reflective coatings (ARCs) have evolved into highly effective reflectance and glare reducing components for various optical materials with wide application in industry. The ARCs for infrared (IR) optical substrates [30] are more demanding because IR materials usually have high index of refraction values, and IR systems require wide spectral transmission wave band, even dual wave band [31]. The key disadvantage of the antireflection coatings is that coating materials are usually soft with low durability. This problem is solved by development of the hard coatings (HC) and diamond-like coatings (DLC) [32,33] suitable for surface protection but with minimal degradation of AR coating properties [34].

#### 2.1.3. Veiling Glares and Ghost Images

Veiling glare is unwanted light in an image arising from reflections and scattering within the system. Such reflections occur from component surfaces, component edges and mounts, sides of barrels, stops and diaphragms, and scattering occurs from component surface defects and contaminants, bulk defects and diffusely reflecting surfaces [35]. Light that is partially reflected from optical boundary window surfaces causes the formation of ghost images. In imaging applications, ghost images may cause contrast reduction, and may veil parts of the image of the nominal scene (veiling glare). Various optical design methods can be applied for ghost light influence minimization [36]. The intensity of a ghost image is proportional to the product of the reflection coefficients of the coatings of the involved refractive interfaces. The reflection coefficients of optical coatings are limited by the currently available technology and can vary depending on the wavelength bandwidth, from 1% or 2% for a simple monolayer coating to about 0.1% for multilayer coatings. The optical protective window is the key optical interface to outer source, so it is important to have high quality antireflection coatings. In addition, the lens input surface should have reflectance that is as low as possible. The potential optical protective window surfaces that can contribute to ghost image formation are illustrated in Figure 5.

In visible light systems, double-reflection ghosts are caused by double-reflection ghost of the sunlight or other dominant illumination source. In an infrared system, a cryogenically cooled detector, having low radiance, acts somewhat like sunlight but generates a dark (low temperature) area in the image. This effect is called the narcissus phenomenon [37,38] and happens due to cold signal being retro-reflected from the lens surface. Actually, this effect is a kind of stray radiation in the IR imager. IR stray light can be minimized using proper design solutions [39]. In addition to image degradation, unwanted IR stray radiation causes changes in background radiation [40] which degrade thermal contrast and increase image noise level. The most effective measure against this is to keep reflection as low as possible on both window surfaces. This goal is not completely achievable because DLC coating, whilst providing high surface durability, has increased reflection.

### 2.2. Physical Effects

Optical protection windows are directly exposed to all environmentally generated influences. Because of this, it is important to examine how the physically generated effects affect overall window performance. Temperature-related effects and mechanical effects could cause optical protective window performance degradation. Aero-optical effects are the best example of how severe physical influences combined with optical window properties may be. Although aero-optical effects could not be found in the common surveillance systems applications, we review them in the Appendix B and refer to the most relevant papers that can be used in the case that more detailed analysis is necessary.

#### 2.2.1. Temperature-Related Effects

Ambient temperature can have influence on window heating, leading to window thermal expansion and change of index of refraction. Window thermal expansion can be easily compensated through proper window mounting design (using an unclamped, simply supported design with O-ring sealing connection to the equipment housing. In the case of a clamped design without lateral translation, window expansion could produce window bending, causing window optical properties to change due to sagittal bending or causing excess stress that leads to a window crash. This case requires additional design analysis and calculation of induced stress that will lead to selection of window materials or redefinition of window thickness.

In IR imaging systems, temperature can cause some additional effects. A window’s high temperature can cause elevated background temperature, thereby increasing the noise level in the image sensor. In addition, non-uniformly heated window material causes complicated optical and mechanical effects at the same time [41,42,43]. Most IR optical materials exhibit a strong index of refraction temperature dependence which can affect the AR coating’s effectivity. IR material emissivity [44,45] can have high value, causing a higher level of background radiation. Low reflection AR coatings have low emissivity, helping to resolve this issue.

Alongside the optical window’s basic function to protect the imager inside housing and act as the so-called “sacrificial element”, the window coupling to equipment housing should allow for easy window replacement; therefore, window mounting and sealing using an O-ring is the best solution. At the same time, this design solution provides the best thermal expansion effects compensation.

In the case of MWIR and LWIR sensors, window heating can have two effects:○homogenous high temperature has influence on the sensor background signal, lowering the imager contrast; it is important to use low-emissivity material and coatings to compensate for this influence.○non-homogenous window heating and/or window materials with an index of refraction high-temperature dependence lead to a non-homogenous index of refraction through optical opening, thereby generating disturbance in the incoming optical imaging wave front. Non-homogenous heating effects require more sophisticated analysis using optical design software and a sensor lens with a detailed layout.

In stationary surveillance systems, excess heating is not expected on one hand, and stationery slow changing ambient temperature will not cause non-homogenous heating on the other hand. That means that optical window temperature effects can be compensated for through related design solutions. In many cases, the application of the solar radiation shield is sufficient to protect window from extreme thermal effects.

#### 2.2.2. Mechanical Effects

In the application environment of the optical protective window, the appearance of external stress is a common source of influence on the optical window. Optical material basic properties such as elasticity, strength (fracture resistivity) and hardness [23,24,25] have influence on a window’s durability against environmental influences. External stress causes internal strain in materials, causing material deformation. In the elastic deformation region (the linear relation between strain and stress), there is the possibility of temporal shape change that can have influence on the window’s optical properties. In the plastic deformation region, there are much complex influences that could cause window fracture.

The strength of an optical material is governed by the random distribution of the size, orientation, and location of the surface flaws or inner material flaws in relation to the regions under stress [46]. Surface flaws are commonly created during the grinding and finishing operations of the optical substrate. In some cases, window processing should be done extremely carefully [47,48]. Due to the scatter in the flaw size, the strength of optical material is a function of the size of the window. Hence, there is no deterministic strength for brittle materials (unless the flaws are extremely uniform). These flaws or cracks propagate under tensile loads to a critical value, and then experience uncontrolled crack growth until the part physically fractures. The optical material fracture mechanism could be developed under three conditions: (a) the existence of initial defects or cracks in the material, caused during material elaboration and machining or created under the effect of an external stress system; (b) the presence of a fracture temperature that is below the critical fragility temperature of material; (c) the presence of internal stress generated during the fabrication process. The initial cracks act as stress concentrators.

In addition, stress propagation inside a material depends on the window mounting condition [48], so a proper mounting design is required in critical applications.

In optical protective windows in surveillance systems, stress causing fractures is rare, but optical distortion caused by external stress is the most common case. There are different definitions of the distortion influences and different methods for their evaluation [49]. The most common definition of distortion is based on tracking rays of light that pass through the window. The second definition describes distortion in terms of local curvature changes, resulting in local focusing or defocusing of light passing through the window. The third definition of distortion is based on a fundamental window attribute, namely changes in the window’s optical path length.

Optical protective windows intended for application in surveillance systems are sensitive to surface damage caused by mechanical effects (scratches) related to the optical material, or coating hardness or durability.

## 3. Optical Protective Window Requirements Definition

Long-range observation systems are designed to operate under harsh environmental conditions while demanding outstanding detection, recognition and identification (DRI) range capabilities, superb image quality, and accurate line-of-sight (LOS). EO/IR windows design is a significant challenge for the long range surveillance system designer who must specify it for high EO performance, durability, producibility and affordable initial and life cycle costs. This is particularly true in the LWIR band, at which window materials and coating choices are limited by the system’s design requirements. The requirements also drive the optimization of numerous mechanical, optical, materials, and electrical parameters. EO/IR imaging system window design is a challenge, as illustrated in Table 1, which presents the interrelationship of the optical, mechanical, and system design processes and their effect on protective window design features.

### 3.1. Optical Requirements

Optical specifications and requirements cover a wide range of needs [50]. Functional specifications related to the image quality or other optical characteristics are required for the satisfactory operation of a protective window, serving as the goal for the design and construction. In addition, these specifications are a basis for tolerances related to protective window design. This may include requirements such as optical materials, a dimensional accuracy spectral range, surface properties, protective and anti-reflective coatings, and so on. Assembly specifications and detailed specifications of optical protective windows can be based upon functional specifications.

There are two types of specifications that must be applied to an optical protective window: mechanical tolerances of the shape that indirectly affect the optical quality, and specialized descriptions related to materials that directly affect the image quality.

#### 3.1.1. Optical Material Selection

The specification of a material requires identification of the material type, and data on the homogeneity class, birefringence, and so on. The method of specifying optical material varies with the material type and manufacturer [50,51]. It is useful to refer to a current catalog or use well-defined standard descriptions to ensure that the correct material quality specification is being used. A condensed review of the optical window material spectral application range is presented in Figure 6. A more detailed review of the physical and optical properties for selected optical materials suitable for protective window design is presented in Appendix A.

The optical material index of refraction and Abbe number, together with the spectral transmission band, define the material basic optical properties.

The general requirements for optical materials are as follows:Requirements for refractive index and dispersion coefficientRequirements for optical uniformityRequirements regarding birefringenceRequirements regarding light absorptionRequirements regarding stripes, bubbles and striae.

In *birefringence*, the refractive index depends on the direction from which the light enters the crystal, and on the light’s polarization. This phenomenon characterizes some crystalline and plastic optical materials.

*Striae* are frozen regions of refractive index non-uniformity which cause lines of refractive index inhomogeneity within the bulk material. Their effect on precision optics is similar to the aberrations. When present in a transmissive optical element, striae can cause a phase shift of the light that passes through it.

*Cracks* and *inclusions* are breaks in the uniformity and continuity of the glass. They cause scattering sites and also interrupt the phase of the light that passes through the glass. Additionally, they may serve as a seed for material crack under stress. All of these flaws should be minimized during the fabrication of precise optical elements, and they should be evaluated during quality assurance testing.

The optical material selected for optical protective windows can be described as optical grade material, meaning that material properties should be suitable for high-precision components, having low tolerance margins for all selected material parameters.

In the design of optical protective windows, one must take care of the following basic material optical properties listed below, although material mechanical and temperature properties are equally important considerations.

The most important properties of optical windows are as follows:Transmittance (external and internal);Surface reflection; andIndex of refraction.

Window total transmission T_w_ is:(2)$$T_{w}=t_{1}\cdot t_{2}\cdot e^{-{\mu t}_{w}}$$
where t_1_ and t_2_ are surface transmission, μ is the material absorption coefficient, and t_W_ is window thickness.
(3)$$t_{1}=t_{2}=1-r$$
where r is surface reflection depending on the material index of refraction, n (see Equation (1)).

A lower index of refraction means lower surface reflection and higher window overall transmittance.

For normal application of a protective optical window, a key requirement is a suitable spectral transmission range. The applicable spectral range of selected materials suitable for optical protective windows is illustrated in Figure 6. Additionally, it is important to have as low as possible surface reflectance in order to increase window transmittance and lower internal multiple reflections of stray light that can lower a scene’s apparent contrast. The solution to this is application of antireflection coating.

#### 3.1.2. Surface Quality

The surface specifications of optical windows affect their optical performance and must be considered when selecting or specifying a window. It is important to make sure your optical window has the appropriate specifications with respect to tightness to meet your application requirements, but over-tolerancing the window will unnecessarily increase the cost.

Surface quality is usually described using surface flatness and irregularity definition, parallelism (wedge) tolerance and cosmetic scratch and dig properties. Surface roughness can be measured absolutely and defined using mean roughness amplitude and maximal peak-to-valley amplitude. Surface flatness is commonly tested by comparing the interference pattern generated when the tested surface is compared with perfectly flat test surface interference. In this case, the analysis of the interference fringe shape and curvature provides enough data to measure the surface accurately. The surface flatness description and definition criteria are presented in Table 2.

Due to the different roles of optical protective windows in surveillance systems, requirements for optical material quality could vary widely, but design requirements derived from image quality requirements are similar. Selected tolerances that can be applied for optical protective windows are presented in Table 3.

*Scratches and digs* are optical protective window surface or coating flaws that can greatly affect the optical performance of a coating. Scratches are lines that are cut along the outer surface of a precision optical element, and digs are tiny pits in the surface. Both can cause scattering sites that limit the transmission through and optical quality of a surface.

#### 3.1.3. Antireflection and Protective Coatings

The thin film coatings (protective and antireflective) that are applied to the optical protective window surfaces require careful consideration. In general, the spectral characteristics, such as passband and maximum reflectivity, need to be specified for an antireflection (AR) coating. Examples of typical AR coating spectral reflection curves for several spectral bands are presented in Figure 7, and typical values for selected standard AR coatings that can be used on optical protective windows in surveillance systems are presented in Table 4.

Requirements for environmental stability also need to be described, with reference to tests for film adhesion and durability. Generally, the optical coatings’ durability specification is the most important and the most difficult task. The coating supplier will have a set of “in-house” specifications that will guarantee a specific result that can be used as the basis for the coatings’ specification. In accordance with the importance of the optical coatings, nowadays, there are standards [52,53,54,55,56,57] defining the optical coatings and making specification process consistent and understandable for any supplier. US military standards [56,57] are still applicable, although they are outdated.

During the optical coating manufacturing process, it is common practice to produce a so-called test (witness) piece or sample.

*Witness samples* are typically flat, 1”-diameter windows that are made of the same material, undergo similar processing, and have identical coatings to the optical protective window. They are created to represent the optical properties of the optical protective window, thereby ensuring optical window quality control (spectral transmission and reflection) and durability of the applied coating.

### 3.2. Physical Requirements

Based on the fracture mechanics analysis, the thickness of the optical window should be determined to ensure the window design is reasonable and reliable.

Considering the complex environment, steady-state temperature fields should be determined and temperature load included in overall force load. The expected deformation of the optical window under the structure–thermal coupling condition should be calculated. In the case that expected deformations are higher than selected critical values, detailed optical analysis should be conducted to predict the impact of the deformation of the optical window on the optical performance of the multi-spectral camera. In most cases of optical protective windows’ application in surveillance, the impact of the optical protective window’s deformation on the imaging quality of the optical system is negligible.

#### 3.2.1. Mechanical Properties

The mechanical strength of the optical protective window depends on the window’s thickness, optical material strength and optical window mounting options. The most common mounting options are presented in Figure 8.

*Modulus of Rupture*, frequently abbreviated to *MOR* (and sometimes referred to as bending strength), is a measure of a specimen’s strength before rupture. It can be used to determine a material sample’s overall strength (unlike the modulus of elasticity, which measures the material’s deflection, but not its ultimate strength).

In the case that the optical window is used for separation between two media with different pressures, the minimal window thickness can be calculated [58], using Equation (4) for circular windows and (5) for rectangular windows:(4)$$t_{W}=0.5\cdot A_{W}\cdot \sqrt{\frac{K_{W}\cdot f_{S}\cdot {\Delta P}_{W}}{S_{F}}}$$
(5)$$t_{W}=0.709\cdot L_{W}\cdot \sqrt{\frac{K_{W}\cdot f_{S}\cdot {\Delta P}_{W}}{S_{F}\cdot \;\left(1+R^{2}\right)}}$$
where
t_W_-window thickness [mm]A_W_-unsupported window area diameter [mm]L_W_, W_W_-window length and widthR-ratio L_W_/W_W_K_W_-support-related empirical coefficient (from 0.75 for a clamped window to 1.25 for an unclamped window)f_S_-safety factor (usually 4 to 6)S_F_-modulus of rupture, expressed in [psi]ΔP_W_-pressure difference, expressed in [psi]

**NOTE**: psi—pound per square inch = 7 kPa = 0.007 MPa; 1 atm = 14.7 psi = 101.324 kPa,

Deflection through a circular window due to pressure difference could have some influence on the window’s optical power, introducing degradation to optical sensor performances. The maximal deflection for a circular window can be calculated [58] using Equation (6).
(6)$$y_{m}=K_{1}\cdot \frac{\Delta P_{W}\cdot {\left(\frac{A_{w}}{2}\right)}^{4}}{E\cdot t_{W}{}^{3}}\;\left[\mathrm{mm}\right]$$
where
t_W_-window thickness [mm]A_W_-unsupported window area diameter [mm]K_1_-empirical coefficient (from 1.71 for a clamped window to 0.696 for an unclamped window)E-Young’s modulus of elasticity (mpa)ΔP_W_-pressure difference (mpa)

Once the window thickness is determined according to the selected clamping and sealing design, it is useful to check window deflection for design conditions. Usually, the deflection value is negligible so the calculated minimal thickness can be used as design thickness; otherwise, we need to increase the thickness until the deflection achieves a satisfactory value. In the worst-case scenario, we need to carry out a detailed optical analysis to optimize the window shape.

#### 3.2.2. Influence of Temperature

Optical protective windows in the stationary surveillance systems are not exposed to aerodynamic heating; usually only solar thermal radiation and ambient temperature contribute to window temperature. Elevated window temperature contributes to additional background noise in infrared thermal imaging channels.

The temperature of optical protective windows should be considered during mounting design selection to avoid additional stress due to different material temperature expansion coefficients. In the case that thermal expansion could cause issues, the un-clamped mounting design is more suitable.

### 3.3. Mechanical Requirements

In the case of a circular optical protective window, one can calculate window size. Using the notations of Figure 2, the minimal diameter of an optical protective window that will provide a proper optical field of view is
(7)$$D_{W}=D_{\mathrm{SA}}+Z_{\mathrm{SW}}\cdot \tan\left(\frac{\theta_{\mathrm{FOV}}}{2}\right)+W_{M}$$
(8)$$D_{\mathrm{OW}}=D_{\mathrm{SA}}+Z_{\mathrm{SW}}\cdot \tan\left(\frac{\theta_{\mathrm{FOV}}}{2}\right)$$

Once the optical protective window is defined (by optical material and coatings, shape, size and thickness) proper mechanical drawing should be produced according to standards applicable to optical components [59,60].

In the case that plastics materials are used, one needs to take care of their specific properties [61].

### 3.4. Optical Protective Window Design Methodology

The design process for optical protective window for application in surveillance systems is illustrated in Figure 9. The starting point is the EO/IR imager’s mission analysis (of environmental stress, pressure difference, temperature profile), followed by surveillance system architecture analysis and a basic performance review of the imaging channel.

Optical protective window design is iterative process that should determine the following factors.

***Window material and coatings***: The imager spectral sensitivity range should be used for initial optical material selection. Final optical material selection should follow window mechanical strength limitations.

***Window thickness:*** Optical protective window thickness should be determined using Equation (4) or (5), and refined after window deflection is checked (Equation (6)).

***Window position*** and ***tilt*** to sensor optical axes should follow the available space in the mechanical envelope and stray light influence analysis.

***Window shape*** and sealing technique should be determined according to application requirements, mechanical envelope and cost.

The design process of an optical protective window should provide proper functioning for application within a surveillance system and allow for easy replacement in case of damage.

## 4. Experimental Results

Practical issues that appeared in our surveillance system application triggered our interest about protective window design and related AR coating application. An example of these issues is the appearance of ghost images when protective window glass without AR coating was applied (shown in Figure 10). Our long-range surveillance systems have built-in window cleaning systems so it was important to have AR coating with proper durability to water soluble cleaning agent.

In Figure 11, a damaged optical protective window is shown; this damage appeared after the sample was exposed to water. The resulting damage was severe for two reasons: (a) the durability requirements of the window were underestimated, and (b) the exposure time to reagent was too long. The problem was solved using a stronger definition of the window’s water solubility requirement.

The second group of experimental results is related to experimental evaluation of the optical protective window’s influence on the imaging system’s performance.

In Figure 12, USAF 1951 test chart images projected by the visible image channel are presented for the same sensor using a protective window with proper AR coatings and without a protective window. There is no visible loss of system resolution.

In Figure 13, we present images of the projected USAF 1951 test chart and related MTF curves for the thermal imaging channel using protective window, and these are compared with those of the imager without an IR protective window. It is visible that the image with a protective window is blurred, causing resolution loss. The protective IR window used in this experiment had an AR layer and DLC coating to provide proper environmental protection. The thermal imager also had a DLC protective layer on the lens outer surface. DLC layers have higher reflection, so the mutual action of a pair of DLC layers causes image blurring. In the case in which we use an IR protective window with a DLC layer, it is not possible to use an IR camera with a DLC protective layer on the lens outer surface. 

The third group of experimental results is related to simulated experiments in laboratory environment to demonstrate how protective window can influence image. In these experiments, we used a protective window with an AR layer only in the central part of the window, as illustrated in Figure 14a.

Experiment 1 is designed per the setup shown in Figure 14a where a visible camera is placed in an enclosure with the aforementioned protective window. The selected test pattern is imaged in the presence of a strong light source outside the camera’s field of view. The related image snapshot is presented in Figure 14b, showing the effects of the optical protective window’s presence and the influence of the protective window’s reflectance.

Experiment 2 is designed per the setup shown in Figure 15a. The test pattern is placed on the bottom of the tube which is covered with a structured optical protective window on the other end. The test pattern is illuminated using an outer wide ceiling light and photographed using a cell phone camera. The related images are presented in Figure 15b. In these images, there are different areas visible. One can clearly distinguish a visible area of reflected ceiling light and a disturbed test pattern image on the part of the optical protective window not covered with an AR layer; this is not visible on the part covered with an AR layer. Additionally, the area of ceiling light disturbed by the cell phone body (i.e., the cell phone shadow) is visible. In that area, there is no disturbance of the test pattern image. The presence of the AR coating has influence on the image’s color appearance.

## 5. Discussion

Experimental results showed that optical protective windows can cause issues and degrade images; images may also be damaged because of their unsuitable design, emphasizing the importance of their proper design.

The selected results of the evaluation of the multi-sensor imaging systems using an optical protective window show that optical protective windows can be designed and applied without noticeable degradation of images; however, they can cause degradation, even when well designed, when they are not properly selected in accordance with the imager’s properties.

The results of these specially designed experiments show some of the effects that optical protective windows can have on the imaging chain, influencing image quality and content.

## 6. Conclusions

The wide application of multi-sensor electro-optical surveillance systems is supported by the successful development and mass production of the focal plane array, or FPA, image sensors. In the majority of applications, key attention was paid to multi-sensor system integration and advanced image processing issues. During this time, more applications were in a harsh environment, requiring application of an optical protective window to isolate imagers from environmental influences. Nowadays, the application of optical protective windows is common in the most deployed systems.

The design and application of optical protective windows in surveillance systems seems to be a relatively easy task, but in real applications it can cause failures that require a great effort to be corrected. These failures show that real solutions need a more rigorous approach that uses multidisciplinary analysis and application of rigid evaluation techniques in the design phase. At the same time, experience and development results from the area of protective aerodynamic domes in missile systems, in which more harsh effects exist, have generated a knowledge base which will aid in better understanding of the effectiveness that an optical protective window can have in delivering proper solutions.

We analyzed the most important processes and influences that could be caused by applying optical protective windows to imaging systems. Using the results of this analysis, we derived a simplified methodology for optical protective window design that can deliver initial window designs. At the same time, we have shown that a rigorous multidisciplinary approach is necessary for full evaluation of optical windows’ performance.

The application of the multidisciplinary knowledge and scientific methodologies in engineering is always necessary for delivering proper design solutions. Good understanding of the processes involved leads to proper selection of technical requirements and easier troubleshooting in the case of failure.

## Figures and Tables

**Figure 1 sensors-23-02784-f001:**
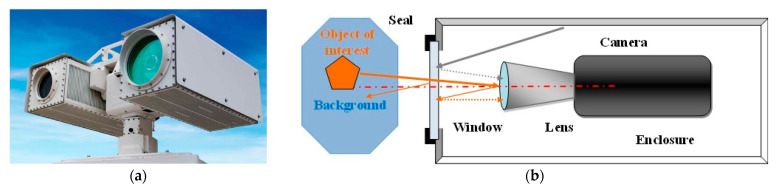
Long range multi-sensor surveillance system: (**a**) typical multi sensor system; (**b**) imaging channel structure.

**Figure 2 sensors-23-02784-f002:**
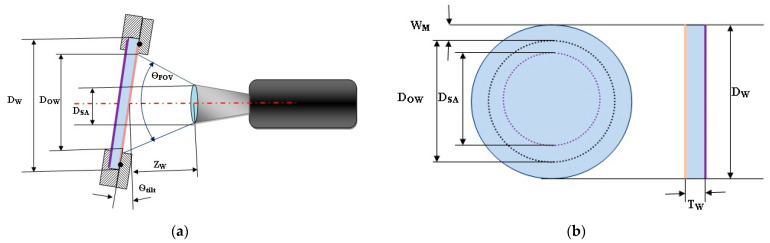
Optical protective window application in multi-sensor surveillance system: (**a**) optical window and imaging channel initial parameters’ definition; (**b**) Optical protective window design parameters.

**Figure 3 sensors-23-02784-f003:**
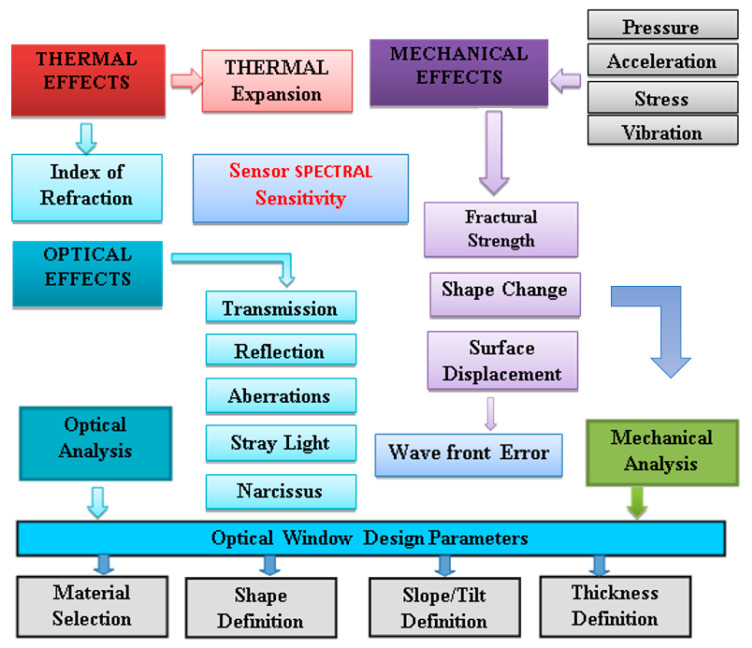
Effects that have influence on optical protective window performances.

**Figure 4 sensors-23-02784-f004:**
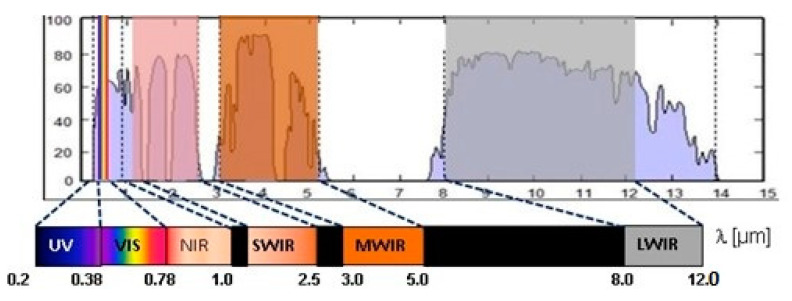
Atmospheric transmission windows and EO/IR sensor spectral sensitivity bands.

**Figure 5 sensors-23-02784-f005:**
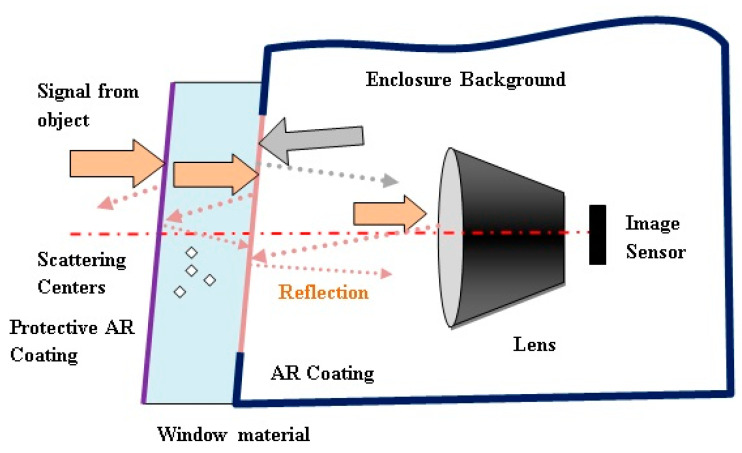
Optical processes related to protective window application.

**Figure 6 sensors-23-02784-f006:**
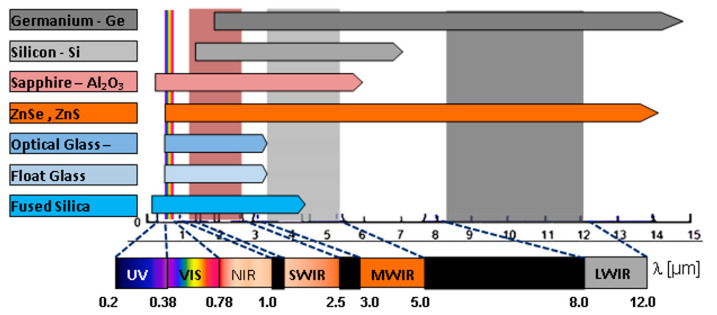
Optical window material spectral application range.

**Figure 7 sensors-23-02784-f007:**
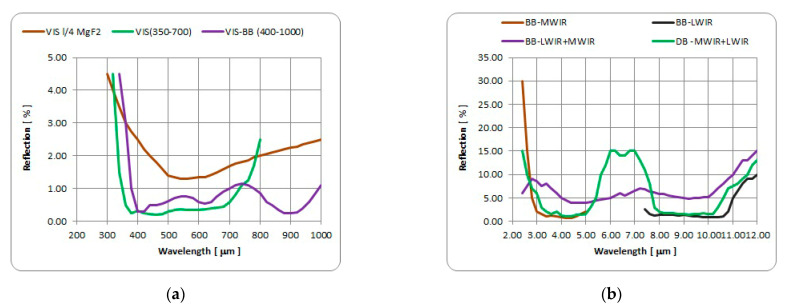
Typical spectral reflection For AR coatings: (**a**) VIS and NIR bands, (**b**) MWIR & LWIR bands.

**Figure 8 sensors-23-02784-f008:**
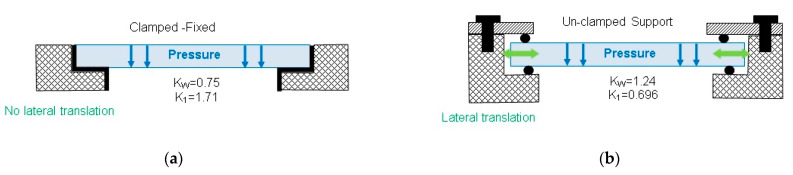
Optical window mounting options: (**a**) Clamped/fixed, and (**b**) Unclamped.

**Figure 9 sensors-23-02784-f009:**
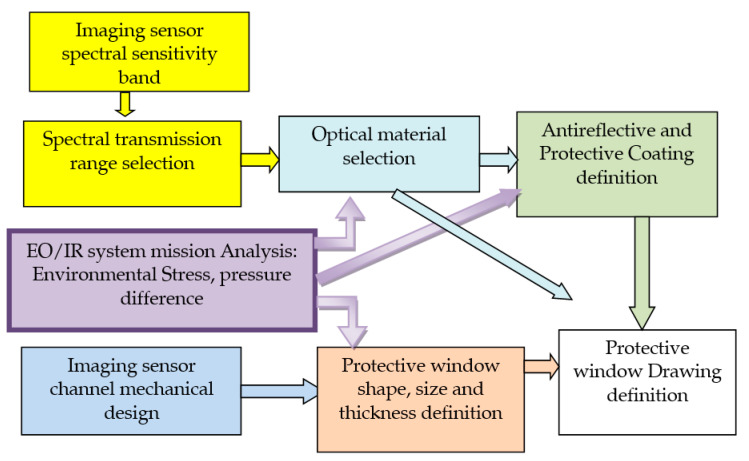
Optical protective window design process (for EO/IR system mission analysis see Figure 3).

**Figure 10 sensors-23-02784-f010:**
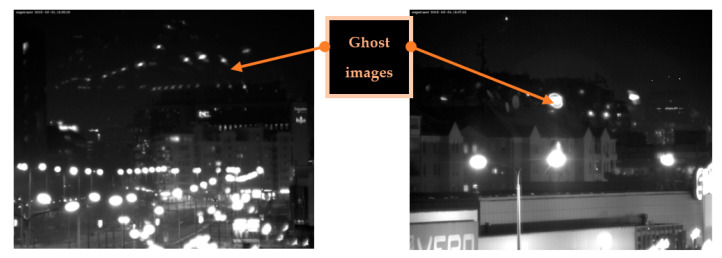
Optical protective window without AR coating (ghost images are visible).

**Figure 11 sensors-23-02784-f011:**
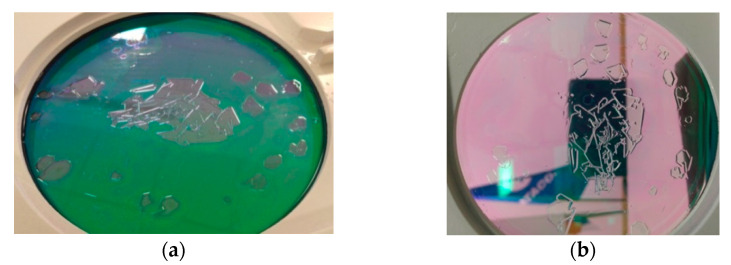
Optical protective window AR coating damage: (**a**) in transmitted light; (**b**) in reflected light.

**Figure 12 sensors-23-02784-f012:**
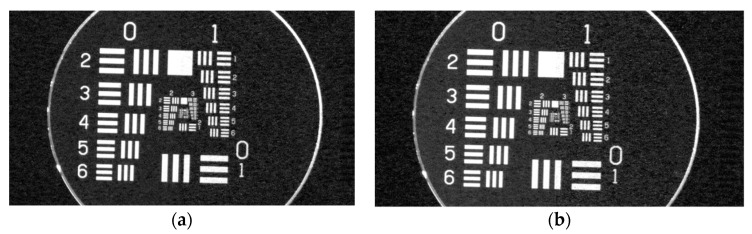
Visible imaging channel projected USAF1951 test chart images: (**a**) without window; (**b**) with protective window.

**Figure 13 sensors-23-02784-f013:**
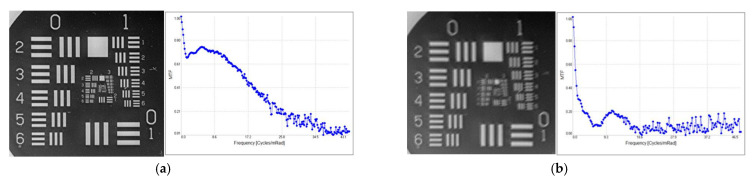
Thermal imaging channel MTF and USAF1951 test chart: (**a**) without window; (**b**) with protective window.

**Figure 14 sensors-23-02784-f014:**
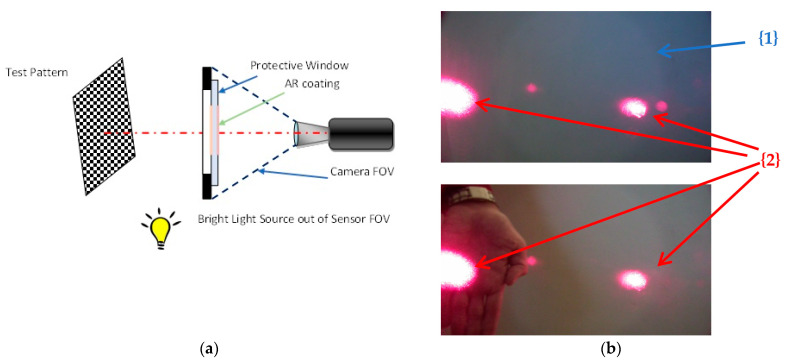
Experiment 1 (**a**) set up (**b**) image snapshots. The top image is an empty scene in which the slightly brighter circle {1} is visible due to an AR layer. The bottom image shows the object in the scene (i.e., the hand); the parasitic reflectance of the outer source {2} is clearly visible and disturbs the scene image.

**Figure 15 sensors-23-02784-f015:**
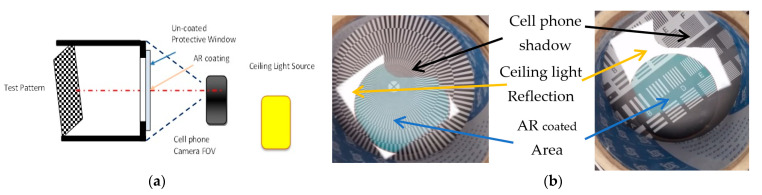
Experiment 2: (**a**) set up; (**b**) image snapshot.

**Table 1 sensors-23-02784-t001:** Optical protective window requirements’ definitions, sources and solutions.

Imager Functional Requirements	Imager Related Window Requirements	Protective Window Design Features
* Spatial-sensitivity & resolution * MRTD, MRC DRI range (target detection, recognition, identification) Resolution Tracking accuracy * Installation/Environment * Operational envelope Structural requirements Rain and dust exposure Window cleaning requirements Maintainability Mission importance and cost	* Sensitivity * Transmission (TV, laser, thermal imager) Emission (window radiance) * Resolution * Window shape deformation Thermal gradients Vibrations Pressure difference Aberration level * Boresight error * Optical distortion * Durability * Rain, sand and dust erosion Chemical agents	* Window substrate material * Spectral transmission band Mechanical properties * Optical quality * Thermal properties * Optical coatings * Antireflection (AR) Erosion resistant (ER) * Position definition * Tilt definition Distance from imager * Size and Shape Definition * Surface finish quality Window frame design Attachment techniques

**Table 2 sensors-23-02784-t002:** Surface flatness description and definition criteria.

*Surface Flatness*	*Description*
≥1 λ	Commonly used for commercial grade applications and in cases in which surface flatness is not critical. ≥1 λ surface flatness is the most cost-effective window option.
λ/4	Used for precision applications in which surface quality is important. This is a common specification for low-to-medium-powered laser systems and high magnification imager windows.
≤λ/10	Used for high-power laser systems and highly precise imaging systems.

**Table 3 sensors-23-02784-t003:** Some general tolerances that can be applied for optical protective windows.

Parameter	Tolerance Guide for Optical Elements		
	*Baseline*	*Precision*	*High Precision*
Diameter	0.1 mm	0.025 mm	0.01 mm
Thickness	0.2 mm	0.05 mm	0.01 mm
Parallelism	5 arc min	1 arc min	15 arc s
Surface irregularity	λ	λ/4	λ/20
Surface finish	5 nm rms	2 nm rms	0.5 nm rms
Scratch/dig	80/50	60/40	20/10
Clear aperture	80%	90%	90%

**Table 4 sensors-23-02784-t004:** A short review of the antireflective coatings’ expected reflection values.

Coating Description (Definition)	Expected Reflection Values
VIS-Single layer—λ/4 MgF_2_	R_min_ < 1.35%, @ λ = 550 nm R_avg_ < 1.7%, @ 400–700 nm
VIS—multi layer—400–700 nm	R_min_ < 0.2%, @ λ = 500 nm R_avg_ < 0.5%, @ 400–700 nm
IS-NIR—Broad band—400–1000 nm	R_avg_ < 1.00%, @ 400–700 nm R_avg_ < 0.85%, @ 800–1000 nm
MWIR (3–5 μm)	R_min_ < 1.35%, @ λ = 3 μm R_avg_ < 3.00%, @ (3–5 μm)
LWIR (8–12 μm)	R_min_ < 1.5%, @ λ = 10 μm R_avg_ < 3.00%, @ (8–12 μm)
MWIR + LWIR—Broad band—BB (3–12 μm)	R_avg_ < 3.00%, @ (3–5 μm) R_avg_ < 3.00%, @ (8–12 μm)
MWIR + LWIR—Dual band—DB (3–12 μm)	R_avg_ < 3.00%, @ (3–5 μm) R_avg_ < 3.00%, @ (8–12 μm)

## Data Availability

Data is contained within the article.

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
