# Peer review of "Optical Protective Window Design and Material Selection Issues in the Multi-Sensor Electro-Optical Surveillance Systems"

_sensors, 2023, doi:10.3390/s23052784_

Round 1

Reviewer 1 Report

The manuscript should be supported by more citations. The main claims should be critically commented on and compared with already published studies. It should be improved.

Author Response

Please find attached the Response to Reviews.

Reviewer 2 Report

Journal: Sensors

Manuscript ID: sensors-2171647 
Title: Optical Protective Window Design and Material Selection Issues in the Multi-Sensor Electro-Optical Surveillance Systems

In this work, the authors proposed simplified methodology and practical recommendation on how to define optical protective window specification in multi-sensor imaging systems using system engineering approach through the analysis of various effects that follow optical window application in connection with imaging systems.

It is an interesting paper and a lot of effort has been done into it. It is well written and of interest to the readers of Sensors. I have the following comments.   

My comments

1.      The novelty of the work is not clear.

2.      What does your work add to the field of multi-sensor imaging systems?  

3.      Please give a new paragraph at the end of the introduction section to illustrate the main aim of this manuscript.

4.      Some of the references are not so recent. To keep the reader updated, I advise the authors to cite some recent references on optical sensing. I suggest the following for example: Opt. Quant. Electron., Vol. 54, No. 2, Article number 127, 17 pages (2022).         Appl. Phys. A 127, 259 (2021).             J. Opt. Soc. Am. B, Vol. 38, No. 8, 2362-2367 (2021).

5.      Figure 9 needs some improvements.  

Author Response

(The authors gave the same response as above.)

Reviewer 3 Report

The paper is a review paper with an important theme.

I have a few comments and suggestions for the authors.

1-     The abstract is very dispersive. The abstract started at line 7 at “..Thorough analysis of various effects…..”

 I suggest rewrite the abstract.

2-     I understand the paper is a review paper, but I suggest the authors work to use a more recent bibliography. Then, the paper will become more useful.

3-     On page 2, the authors said,”.. One of the goals of this article…” what about the others?

4-     Below the authors said, “…. The key objective of this paper is to give a review of the most important….”

I think there is a lack of connection in the paper. The authors could rewrite the text about goals and key objectives.

5-     Sometimes the authors use empty bullets and sometimes fulfilled bullets, as seen on page 3.

6-     The terms SWIR, MWIR and LWIR must be written correctly since they are acronyms.

7-     I suggest the author take care with the use of figures from different authors.

8-     Figure 3 must be improved since there are arrows that do not bring clear information for the readers.

9-     On page 15 I suggest the authors use WL instead Lw and Wt instead tW.

10-  10 At the Experimental Results, I suggest the authors show the setup and inform the used equipment.

11-  The word setup appears a few times as set up.

12-  On  the Authors' contribution I suggest the name of the authors appear in the bibliography citation form.

13-  References are not in the same format. Compare pleas reference 48 and reference 67 as an example.

Author Response

(The authors gave the same response as above.)

Round 2

Reviewer 1 Report

The manuscript has been improved.

Reviewer 2 Report

In my opinion, it can be published in the present form. 

Reviewer 3 Report

Accept in present form since the authors did all the asked changes.